# Greater thermoregulatory strain in the morning than late afternoon during judo training in the heat of summer

Hidenori Otani[1]*, Takayuki Goto[2], Yuki Kobayashi[2], Minayuki Shirato[3], Heita Goto[4], Yuri Hosokawa[5], Ken Tokizawa[6], Mitsuharu Kaya[7]

1 Faculty of Health Care Sciences, Himeji Dokkyo University, Himeji, Hyogo, Japan, 2 National Institute of Technology, Akashi College, Akashi, Hyogo, Japan, 3 Meiji Gakuin University, Tokyo, Japan, 4 Kyushu Kyoritsu University, Kitakyushu, Fukuoka, Japan, 5 Faculty of Sport Sciences, Waseda University, Tokorozawa, Saitama, Japan, 6 National Institute of Occupational Safety and Health, Kiyose, Tokyo, Japan, 7 Hyogo University of Health Sciences, Kobe, Hyogo, Japan

* hotani@himeji-du.ac.jp

**Data Availability Statement:** All relevant data are within the manuscript and its Supporting Information files.

## Abstract

### Purpose

The time-of-day variations in environmental heat stress have been known to affect thermoregulatory responses and the risk of exertional heat-related illness during outdoor exercise in the heat. However, such effect and risk are still needed to be examined during indoor sports/exercises. The current study investigated the diurnal relationships between thermoregulatory strain and environmental heat stress during regular judo training in a judo training facility without air conditioning on a clear day in the heat of summer.

### Methods

Eight male high school judokas completed two 2.5-h indoor judo training sessions. The sessions were commenced at 09:00 h (AM) and 16:00 h (PM) on separate days.

### Results

During the sessions, indoor and outdoor heat stress progressively increased in AM but decreased in PM, and indoor heat stress was less in AM than PM (mean ambient temperature: AM 32.7±0.4°C; PM 34.4±1.0°C, $P<0.01$). Mean skin temperature was higher in AM than PM ($P<0.05$), despite greater dry and evaporative heat losses in AM than PM ($P<0.001$). Infrared tympanic temperature, heart rate and thermal sensation demonstrated a trial by time interaction ($P<0.001$) with no differences at any time point between trials, showing relatively higher responses in these variables in PM compared to AM during the early stages of training and in AM compared to PM during the later stages of training. There were no differences between trials in body mass loss and rating of perceived exertion.

**Funding:** H.O. Grant number 19K11513 Japan Society for the Promotion of Science https://www.jsps.go.jp/j-grantsinaid/index.html NO.

**Competing interests:** The authors have declared that no competing interests exist.

## Conclusions

This study indicates a greater thermoregulatory strain in the morning from 09:00 h than the late afternoon from 16:00 h during 2.5-h regular judo training in no air conditioning facility on a clear day in the heat of summer. This observation is associated with a progressive increase in indoor and outdoor heat stress in the morning, despite a less indoor heat stress in the morning than the afternoon.

## Introduction

A greater thermoregulatory strain (i.e. higher body temperature and heart rate [HR]) has been reported in the morning exercise session from 09:00 h than in the late afternoon exercise session from 16:00 h in high school athletes during 3-h moderate-intensity baseball training [1] and 2-h high-intensity football training [2] in the heat outdoors under a clear sky. Given that there were no time-of-day differences in ambient temperature ($T_a$) and wet-bulb globe temperature (WBGT) between the sessions, these observations occurred because of the differences in environmental heat stress during exercise which increased with rising solar radiation and elevation angle in the morning but decreased with falling solar radiation and elevation angle in the late afternoon [1, 2]. Otani and colleagues [1, 2] therefore concluded that 2–3 h moderate-to high-intensity exercise in the heat of summer under a clear sky may be at a relatively higher risk for developing exertional heat-related illness in the morning from 09:00 h than in the late afternoon from 16:00 h. Those conclusions indicate that the diurnal variations in environmental heat stress affect thermoregulatory responses and the risk of exertional heat-related illness during outdoor exercise in a hot environment. However, such effect and risk are still needed to be examined during indoor sports/exercises.

Judo is a popular combat sport in junior high school and high school athletics in Japan. Judo has been reported to have one of highest numbers of exertional heat-related illness among school organized sports activities in Japan [3]. This is possibly due to a luck of air conditioning in the most judo facilities owing to its high running costs. Hence, majority of judo training sessions during the summer are performed under severe heat stress conditions, including high $T_a$ and relative humidity (RH). This means that an increase in outdoor heat stress raises indoor heat stress in a judo facility in the heat of summer. Sport-specific characteristics of judo may also be responsible for increasing the risk of heat-related illnesses, since judo has high-density efforts and is a high-intensity sport [4] that requires a high-level of strength and endurance performance [5]. Moreover, there are seven weight divisions in judo and athletes are generally required to lose weight before competitions which may induce cumulative dehydration or hypohydration. Also, heavier weight class judokas have high body mass index which produces more metabolic heat and is less efficient in dissipating heat during exercise [6]. Both dehydration/hypohydration and high body mass index have been recognised as a thermoregulatory challenge and a higher risk for developing exertional heat-related illness during exercise in the heat [6]. To our knowledge, only one study has reported the impact of regular judo training on physiological responses in no air conditioning facility in the summer [7]. However, the study [7] was limited to the assessment of hydration status during 90 min regular judo training under a moderate heat stress (29.5°C $T_a$). Therefore, no study has investigated the effect of time-of-day changes in indoor and outdoor heat stresses on thermoregulatory responses during regular judo training in a facility without air conditioning in the heat of summer.

The aim of the current study was therefore to investigate the diurnal relationships between thermoregulatory responses and indoor and outdoor heat stresses during regular judo training in a judo training facility without air conditioning in the heat of summer. We hypothesised that thermoregulatory strain during the training would be greater in the morning than in the late afternoon due to a progressive increase in heat stress in both indoor and outdoor environments during the morning.

## Methods

### Participants

Participants were eight healthy, heat-acclimatized males who belonged to a high school judo team (mean±standard deviation [SD]; age 16.5±1.0 y, height 167±6 cm, body mass 66±11 kg, BMI 23±4 kg·m$^{-2}$, years of training 6±2 y). All data collections were completed in August to ensure that participants were naturally acclimated to the heat. Their weight divisions were 2 extra lightweight, 1 half lightweight, 2 lightweight, 2 half middleweight and 1 middleweight. They trained ~5 days per week and performed a similar protocol of training in the current study more than 12 weeks. All participants and their parents received written information regarding the nature and purpose of this study prior to participation in the study. Following an opportunity to ask questions, a written statement of consent was signed by their parents. The protocol employed was approved by the local Ethics Advisory Committee of Himeji Dokkyo University (REF: 19–05) and was conducted in accordance with principles of the Declaration of Helsinki.

### Experimental protocol

All participants completed two 2.5-h regular judo training sessions in a judo facility. A building was located approximately 15 m from the north side of the facility; however, there were no obstructions to shield the sun within a 50 m radius from the east, south and west sides of the judo facility. The judo facility was a one-story building with a floor space of 225 m$^2$ (15 m × 15m). The judo facility had no air conditioning and there were windows on east and west sides of the wall, which were kept open during the sessions. The sessions were commenced at two different times-of-day: 09:00 h (AM) and 16:00 h (PM). The present study was conducted in early-August on a completely clear day, and PM trial was conducted first and AM trial was carried out two days later. A normal training session took place two days before the first trial (PM trial) but no exercise was permitted during the 24 h prior to the trials. Participants were dressed in the same judo uniform (jacket, pants, belt) in both trials. In the current study, participants wore a T-shirt under the judo uniform to protect surface skin temperature thermistor probes at the chest and upper arm during the sessions. This judo ensemble was 2.5±0.1 kg of total weight. There were no studies reporting the intrinsic clothing insulation ($R_{cl}$), the clothing area factor ($f_{cl}$) and the evaporative resistance of clothing ($R_{e,cl}$) of judo uniform. The present study therefore used the similar clothing to estimate $R_{cl}$, including 0.18 clo of short sleeve, sport shirt, 0.50 clo of double-breasted suit, jacket (denim), and 0.32 clo of straight, long, loose (denim) [8]. A clo is a unit of thermal insulation for clothing: one clo can be defined as the amount of insulation that allows the transfer of 1 W·m$^{-2}$ with a temperature gradient of 0.155˚C between two surfaces (0.18˚C·m$^2$·h·kcal$^{-1}$). Total $R_{cl}$ was calculated as 0.77 × the sum of these $R_{cl}$ [8]. The $f_{cl}$ was calculated as (0.305 × total $R_{cl}$) + 1.0 [8]. As a result, total $R_{cl}$ was 0.770 clo or 0.119 W·(m$^2$·˚C)$^{-1}$ and $f_{cl}$ was 1.23. Since this total $R_{cl}$ was similar to that of baseball uniform-temperate weather (0.762 clo) and football uniform-warm weather (0.795 clo) [9], the current study used $R_{e,cl}$ of these uniforms which were 0.022 W·(m$^2$·kPa)$^{-1}$.

Participants entered a laboratory which was close to the judo facility after a 2 h fast in each trial with the exception of plain water, which was allowed until 30 min before the start of the trials. Upon arrival, participants first emptied their bladder and thereafter nude body mass was measured to the nearest 10 g (AD6205B, A&D Co., Ltd., Tokyo, Japan). Surface skin temperature thermistor probes (ITP082-25, Nikkiso-Therm Co., Ltd., Musashino, Tokyo, Japan) were attached to four sites (chest, upper arm, thigh and calf) under the clothing without preventing range of motion. A weighted average of chest (0.3), upper arm (0.3), thigh (0.2) and calf (0.2) skin temperatures was used to calculate mean skin temperature ($T_{sk}$) [10]. Gastrointestinal thermometry has been shown to be a valuable device for core temperature ($T_{core}$) assessment in the field and athletics settings [11]. However, the current study measured an infrared tympanic temperature ($T_{ty}$) to estimate $T_{core}$ due to the restriction from pharmaceutical affairs law in Japan using gastrointestinal thermometry. $T_{ty}$ was measured using an infrared tympanic thermometer (GeniusTM 2, Covidien, Mansfield, MA, USA). In each measurement, two consecutive readings were obtained. All $T_{ty}$ measurements were taken by a single operator, using the recommended technique [12]. To avoid the increased effects of increasing $T_a$ on $T_{ty}$ in the heat [13], the thermometer was stored inside a cooling box during the trial. The temperature inside this box was maintained by ice packs at about 25°C. Thermal sensation (TS) was measured using a 9-point scale [14]. All pre-exercise measurements were carried out in the laboratory in a temperate environment (25–27°C $T_a$) because prior heat stress exposure may increase thermoregulatory strain during subsequent exercise-heat stress in the morning than in the afternoon [15].

Participants then entered the judo facility and commenced a 2.5-h training session. Participants started the sessions in a dry judo uniform. Participants received airflow during the sessions which was directed from 3 corners to the centre of the facility by 3 industrial fans, using a 0.5 m blade diameter fan (SF-50FS-1VP, Suiden Co. Ltd., Sangocho, Osaka, Japan), to prevent high levels of hyperthermia during exercise in the heat [16]. Mean air velocity in the judo facility was about 2.5 km·h$^{-1}$. Both training sessions were led by the same judo instructor to retain consistency between the two experiments. The content of training session in both trials was as follows: warm-up (20 min: running, dynamic-stretching and ukemi); newaza randori (25 min); rest (5 min); tachiwaza uchikomi (20 min); nagekomi (10 min); tachiwaza randori (40 min); waza practice (5 min); rest (5 min); strength training-own weight (15 min); and cool-down (5 min: static-stretching) (Fig 1). During the sessions, $T_{ty}$ and TS were assessed every 60 min and at the end of the training (Fig 1). To determine whole-body perception of effort, rating of perceived exertion (RPE) was assessed every 60 min and at the end of the training using the 6–20 RPE scale [17] (Fig 1). Skin temperatures (thermometer N543R, Nikkiso-Therm Co., Ltd., Musashino, Tokyo, Japan) and HR (HR monitor A370, Polar Electro, Kempele, Finland) were also recorded every 60 min and at the end of the training (Fig 1). Participants were free to ingest plain water maintained at about 30°C during the sessions. Following the sessions, participants returned to the laboratory, removed the probes and re-measured nude body mass to allow the estimation of total sweat loss.

## Environmental measurements

Environmental conditions were measured at both inside and outside of the judo facility. Indoor environmental conditions were measured 1.5 m above the floor. Outdoor environmental conditions were measured 1.5 m above a dark asphalt pavement close to the judo facility. $T_a$, RH, black globe temperature ($T_g$), and WBGT were measured using a WBGT meter (WBGT-203A; Kyoto Electronics Industry Co., Ltd., Fukuchiyama, Kyoto, Japan) every 30 min. Air velocity was measured using an anemometer (AM-4214SD; Mother Tool Co., Ltd.,

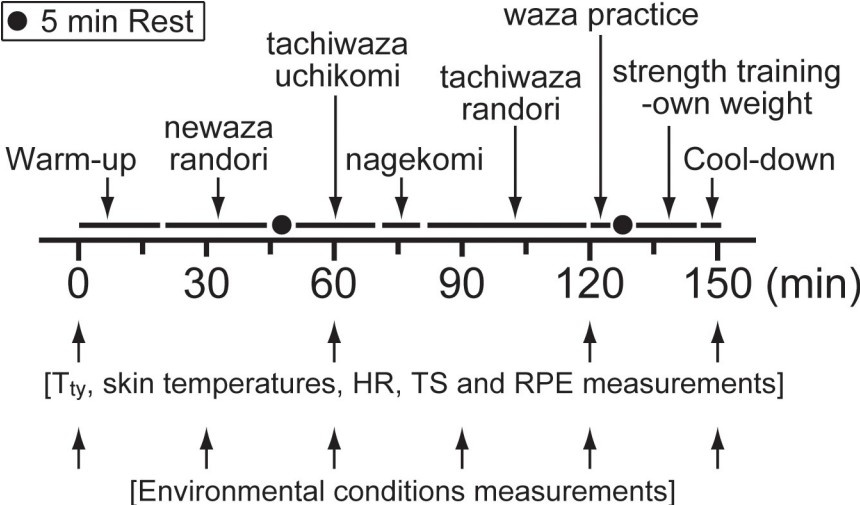

**Fig 1. Schematic representation of the experimental protocol.** $T_{ty}$, infrared tympanic temperature; HR, heart rate; TS, thermal sensation; RPE, rating of perceived exertion.

Ueda, Nagano, Japan) facing the headwind every 30 min. Direct and diffuse solar radiation in the horizontal plane was recorded using a pyranometer (MS-01; Eko Instruments Co., Ltd., Tokyo, Japan) every 30 min, and solar radiation (global) was estimated by summing up the values.

## Calculations

The equations of $T_{sk}$, mean radiant temperature ($T_r$), dry or sensible heat loss (DHL), evaporative heat loss (EHL), total heat loss (THL), absolute humidity, total sweat loss and age-predicted maximal HR (HRmax) are included in supporting information (S1 Data).

## Statistical analyses

Data are presented as mean±SD. The significance level was set at $P<0.05$. The normality of the data and the homogeneity of variance between the trials were tested using Shapiro-Wilk's test and Levene's test, respectively. Non-parametric data (TS) were analysed using R (version 4.0.2). TS was analysed using a two-way (time-of-day [two levels, i.e., AM and PM] × time [four levels, i.e., 0, 60, 120 and 150 min]) repeated measures ANOVA with the R package nparLD (version 2.1) for the LD-F2 design. Pair-wise differences between trials were evaluated using the Tukey multiple comparison tests. In all other cases, statistical analyses of data were done in the IBM SPSS (version 21; IBM Corp., Armonk, N.Y., USA). Data collected once per trial were analysed using a one-way repeated measures ANOVA, and data collected over time were analysed using a two-way (time-of-day [two levels, i.e., AM and PM] × time [three or four levels, i.e., 0, 60, 120 and 150 min]) repeated measures ANOVA. Pair-wise differences between trials were evaluated using one-way ANOVAs with a Bonferroni adjustment applied for multiple comparisons. Environmental parameters were analysed using the independent (AM vs. PM) and dependent (indoor AM vs. outdoor AM; indoor PM vs. outdoor PM) samples t-test. Cohen's d ($d$) was used as a measure of effect size for parametric paired samples; a $d$ of 0.2 to $<0.5$ and $\geq 0.5$ to $<0.8$ has been suggested to represent a small and medium treatment effect, respectively, while a $d \geq 0.8$ represents a large treatment effect [18]. Spearman's rank correlation coefficient ($r_s$) was used to assess the relationship between the changes in $T_{ty}$, $T_{sk}$

and HR in each subject and the changes in $T_a$, WBGT, $T_g$, $T_r$, and solar radiation at 60, 120 and 150 min. A $r_s$ of <0.2 were considered a weak correlation, 0.21–0.4 were considered fair, 0.41–0.6 were regarded as moderate, 0.61–0.8 were deemed strong and 0.81–1.0 very strong [19].

## Results

Pre-exercise body mass ($P = 0.580$), $T_{sk}$ ($P = 0.349$) and HR ($P = 0.247$) were not different between trials, but pre-exercise $T_{ty}$ was higher on PM than AM trial ($P<0.001$; $1−β = 1.00$; $d = 1.51$; Fig 2A).

### Environmental conditions

In indoor environmental conditions, $T_a$ ($1−β = 0.97$; $d = 2.23$) and $T_g$ ($1−β = 0.88$; $d = 1.74$) were lower and RH ($1−β = 0.66$; $d = 1.27$) was higher on AM than PM trial (Table 1). In out-door environmental conditions, absolute humidity ($1−β = 0.96$; $d = 2.15$) and solar radiation ($1−β = 1.00$; $d = 3.19$) were higher on AM than PM trial (Table 1).

In AM trial, higher RH ($P<0.05$; $1−β = 0.90$; $d = 1.81$) and absolute humidity ($P<0.05$; $1−β = 0.89$; $d = 1.79$) and lower WBGT ($P<0.01$; $1−β = 0.88$; $d = 1.75$), $T_g$ ($P<0.001$; $1−β = 1.00$; $d = 10.05$) and $T_r$ ($P<0.001$; $1−β = 1.00$; $d = 5.19$) were apparent on the indoor than outdoor environmental conditions. In PM trial, higher $T_a$ ($P<0.001$; $1−β = 0.99$; $d = 2.54$), absolute humidity ($P<0.01$; $1−β = 1.00$; $d = 2.91$) and WBGT ($P<0.05$; $1−β = 0.41$; $d = 0.88$) and lower $T_r$ ($P<0.05$; $1−β = 0.93$; $d = 1.95$) were observed on the indoor than outdoor environmental conditions.

### Body temperature responses

There was a trial by time interaction effect for $T_{ty}$ ($P<0.05$; $1−β = 0.82$), but post hoc analysis revealed no difference at any time point between trials (all $P>0.05$; Fig 2A). Also, no main effect of trial was observed in $T_{ty}$ ($P = 0.137$). Although no interaction ($P = 0.065$) was shown in $T_{sk}$ between trials, there was a main effect of trial in $T_{sk}$ ($P<0.05$; $1−β = 0.53$) which was higher on AM than PM trial ($P<0.05$; $d = 0.22$: Fig 2B).

### Heart rate response

A trial by time interaction effect was detected for HR ($P<0.05$; $1−β = 0.91$), but with post hoc adjustment there was no difference at any time point between trials (all $P>0.05$; Fig 2C). The percentage of HRmax (% HRmax) at 60, 120 and 150 min was 67±6%, 67±6% and 55±3% in AM trial and 72±7%, 63±4% and 57±7% in PM trial. There was a trial by time interaction effect for % HRmax ($P<0.05$; $1−β = 0.87$), but post hoc analysis revealed no difference at any time point between trials (all $P>0.05$). The average HR during exercise was not different between trials (AM 63±4% HRmax, PM 64±5% HRmax; $P = 0.680$).

### Heat loss responses

DHL (AM 5.3 W·m$^{-2}$, PM −7.0 W·m$^{-2}$; $d = 7.79$), EHL (AM 104.0 W·m$^{-2}$, PM 97.9 W·m$^{-2}$; $d = 1.92$) and THL (AM 109.3 W·m$^{-2}$, PM 90.9 W·m$^{-2}$; $d = 3.89$) were greater on AM than PM trial (all $P<0.001$; all $1−β = 1.00$; Fig 3).

### Body fluid balance

There were no differences between trials in the volume of water ingested (AM 2254±718 mL; PM 2269±740 mL: $P = 0.905$), body mass loss (AM 1.2±0.6%; PM 1.3±0.9%: $P = 0.565$), total

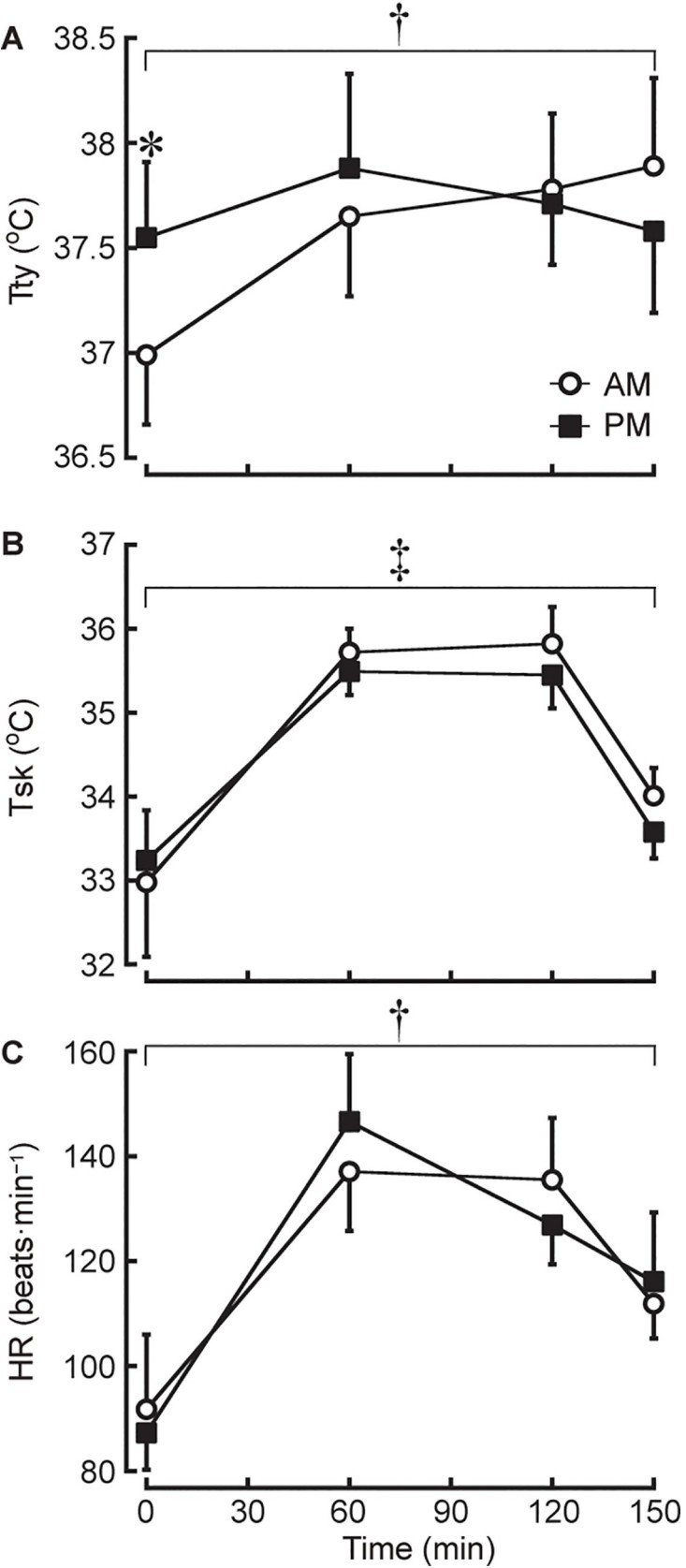

**Fig 2.** Changes in infrared tympanic temperature ($T_{ty}$; A), mean skin temperature ($T_{sk}$; B) and heart rate (HR; C) during exercise. $^*P<0.05$ denotes a difference of pre-exercise between AM and PM trials. $†P<0.05$ denotes an interaction between AM and PM trials. $‡ P<0.05$ denotes a main effect of trial between AM and PM trials.

sweat loss (AM 3.1±0.8 kg; PM 3.1±0.7 kg: $P = 0.509$) and sweat rate (AM 1.23±0.30 L/h; PM 1.26±0.27 L/h: $P = 0.481$).

## Perceptual responses

A trial by time interaction effect was shown for TS ($P<0.05$), but post hoc analysis revealed no difference at any time point between trials (all $P>0.05$; Fig 4A). Also, there was no main effect of trial in TS ($P = 0.137$). There was no interaction ($P = 0.214$) and main effect of trial ($P = 0.089$) in RPE, although a tendency was observed in a main effect of trial (Fig 4B).

## Relationship between the changes in $T_{ty}$, $T_{sk}$ and HR and in environmental conditions

The changes in Tty was not correlated with the changes in environmental conditions in both trials. The changes in $T_{sk}$ and HR in PM trial was strongly correlated with the changes in $T_a$, $T_g$, WBGT and $T_r$ in the indoor environment and $T_g$, WBGT, $T_r$ and solar radiation in the outdoor environment (Table 2). Also, the changes in HR in PM trial was moderately correlated with the changes in $T_a$ in the outdoor environment.

## Discussion

The current results demonstrated a higher $T_{sk}$ in AM than PM trial (Fig 2B), despite greater DHL and EHL in AM than PM trial (Fig 3) with no differences between trials in hydration status. $T_{ty}$ and HR showed a trial by time interaction effect with no differences at any time point between trials (Fig 2A and 2C), indicating that these were relatively higher in PM than AM trial at the early stages of training but in AM than PM trial at the later stages of training. A novel finding in this study is that there is a greater thermoregulatory strain in the morning from 09:00 h than in the late afternoon from 16:00 h during 2.5-h regular judo training in a judo training facility without air conditioning in the heat of summer. This finding is associated with the progressive increase in indoor and outdoor heat stresses in the morning compared with the progressive decrease in indoor and outdoor heat stresses in the late afternoon. Therefore, our experimental hypothesis was confirmed. These findings are consistent with that of Otani and colleagues which involved 3-h moderate-intensity baseball training [1] and 2-h high-intensity football training [2] in high school athletes in the heat outdoors under a clear sky. As concluded in the previous studies [1, 2], the present study indicates that an increase in indoor heat stress during AM trial may cause a greater thermoregulatory strain than a decrease in indoor heat stress during PM trial, regardless of a smaller indoor heat stress in AM than PM trial and no solar radiation effect on both trials. Hence, the current study supports the previous studies [1, 2] and the risk for developing exertional heat-related illness during exercise may be relatively higher in the morning from 09:00 h than in the late afternoon from 16:00 h when 2.5-h regular judo training is performed in a judo training facility without air conditioning on a clear day in the heat of summer.

Regarding the relationships between indoor heat stress and physiological responses in no air conditioning facility during regular judo training in the summer, only one study of Revera-Brawn & Félix-Dávila [7] has reported the changes in hydration status in adolescent judokas during 90 min judo training in the afternoon from 15:30 h in a temperate environment

**Table 1. Indoor (Judo facility) and outdoor environmental conditions during each trial.**

| | | | Time | | | | Mean ± SD | p value |
|---|---|---|---|---|---|---|---|---|
| | 0 | 30 | 60 | 90 | 120 | 150 | | |
| Indoor environment | | | | | | | | |
| Ta, °C | | | | | | | | |
| AM | 32.0 | 32.4 | 32.8 | 33.0 | 33.0 | 33.1 | 32.7 ± 0.4 | 0.008 |
| PM | 35.5 | 35.3 | 34.8 | 34.1 | 34.0 | 32.4 | 34.4 ± 1.0 | |
| RH, % | | | | | | | | |
| AM | 64 | 62 | 61 | 62 | 61 | 59 | 61 ± 2 | 0.037 |
| PM | 60 | 52 | 52 | 58 | 56 | 62 | 57 ± 4 | |
| AH, g·m$^{-3}$ | | | | | | | | |
| AM | 21.7 | 21.4 | 21.5 | 22.1 | 21.6 | 21.2 | 21.6 ± 0.3 | 0.857 |
| PM | 24.4 | 20.9 | 20.4 | 21.9 | 21.1 | 21.4 | 21.7 ± 1.3 | |
| AV, km·h$^{-1}$ | | | | | | | | |
| AM | 2.5 | 2.5 | 2.5 | 2.5 | 2.5 | 2.5 | 2.5 ± 0.0 | 1.000 |
| PM | 2.5 | 2.5 | 2.5 | 2.5 | 2.5 | 2.5 | 2.5 ± 0.0 | |
| WBGT, °C | | | | | | | | |
| AM | 28.8 | 28.9 | 29.0 | 29.3 | 29.1 | 29.4 | 29.1 ± 0.2 | 0.074 |
| PM | 31.6 | 30.1 | 29.7 | 30.0 | 29.5 | 28.8 | 29.9 ± 0.8 | |
| Tg, °C | | | | | | | | |
| AM | 34.5 | 35.0 | 35.0 | 34.9 | 35.1 | 36.6 | 35.2 ± 0.7 | 0.021 |
| PM | 38.3 | 37.5 | 37.2 | 36.8 | 36.4 | 34.6 | 36.8 ± 1.1 | |
| Tr, °C | | | | | | | | |
| AM | 38.7 | 39.5 | 38.9 | 38.0 | 38.6 | 42.4 | 39.4 ± 1.4 | 0.114 |
| PM | 42.8 | 41.4 | 41.2 | 41.4 | 40.4 | 38.2 | 40.9 ± 1.4 | |
| Outdoor environment | | | | | | | | |
| Ta, °C | | | | | | | | |
| AM | 31.9 | 31.9 | 32.6 | 32.9 | 33.0 | 33.6 | 32.7 ± 0.6 | 0.119 |
| PM | 33.5 | 32.9 | 31.5 | 31.0 | 30.7 | 30.2 | 31.6 ± 1.2 | |
| RH, % | | | | | | | | |
| AM | 58 | 58 | 57 | 58 | 58 | 60 | 58.2 ± 0.9 | 0.416 |
| PM | 53 | 55 | 58 | 58 | 59 | 60 | 57.2 ± 2.4 | |
| AH, g·m$^{-3}$ | | | | | | | | |
| AM | 19.5 | 19.5 | 19.9 | 20.6 | 20.7 | 22.1 | 20.4 ± 0.9 | 0.008 |
| PM | 19.4 | 19.5 | 19.1 | 18.6 | 18.6 | 18.4 | 18.9 ± 0.4 | |
| AV, km·h$^{-1}$ | | | | | | | | |
| AM | 3.0 | 5.0 | 5.0 | 2.0 | 10.0 | 7.0 | 5.3 ± 2.6 | 0.469 |
| PM | 7.0 | 6.0 | 4.5 | 4.0 | 2.0 | 2.0 | 4.3 ± 1.9 | |
| WBGT, °C | | | | | | | | |
| AM | 29.0 | 29.3 | 29.8 | 30.2 | 30.3 | 31.2 | 30.0 ± 0.7 | 0.147 |
| PM | 31.0 | 30.3 | 28.6 | 28.2 | 28.4 | 27.7 | 29.0 ± 1.2 | |
| Tg, °C | | | | | | | | |
| AM | 42.0 | 42.5 | 43.0 | 43.5 | 43.8 | 44.8 | 43.3 ± 0.9 | 0.058 |
| PM | 45.0 | 43.2 | 38.5 | 38.1 | 37.2 | 35.5 | 39.6 ± 3.4 | |
| Tr, °C | | | | | | | | |
| AM | 58.7 | 65.5 | 65.6 | 57.3 | 77.3 | 73.2 | 66.3 ± 7.2 | 0.124 |
| PM | 74.1 | 67.8 | 53.9 | 52.7 | 46.4 | 43.1 | 56.3 ± 11.1 | |
| SR, W·m$^{-2}$ | | | | | | | | |

(*Continued*)

**Table 1.** (Continued)

| | | | Time | | | | Mean ± SD | p value |
|---|---|---|---|---|---|---|---|---|
| | 0 | 30 | 60 | 90 | 120 | 150 | | |
| AM | 860 | 920 | 930 | 1020 | 1060 | 1100 | 982 ± 85 | <0.001 |
| PM | 810 | 510 | 300 | 220 | 160 | 110 | 352 ± 242 | |

Ta, ambient temperature. RH, relative humidity. AH, absolute himidity. AV, air velocity

WBGT, wet-bulb globe temperature. Tg, black globe temperature. Tr, mean radiant temperature. SR, solar radiation.

(29.5˚C $T_a$). Their study [7] showed that body mass loss, sweat rate and the volume of water ingested during the training were 1.9±0.5%, 0.8±0.3 L/h and 257±246 mL, respectively. These results indicate much greater body mass loss and an excessively lower sweat rate compared with the present results. These disagreements may exist due to the low volume of water ingested and an about 3–5˚C lower $T_a$ in the previous study [7] compared to the current study. Moreover, no information about exercise intensity was reported in the previous study [7]. Besides, only one study [20] reported the changes in $T_{ty}$ during regular judo training whilst wearing a cooling vest could have attenuated a greater increase in $T_{ty}$ during 65 min regular judo training and 10 min post-training recovery. The authors reported that the temperature was 27˚C $T_a$ at the beginning of training but they did not state the location where the temperature was measured (indoor or outdoor) and a presence of an air conditioner in the facility [20]. Considering that judo is the most popular combat sport in the world as well as Japanese junior high and high schools, more research is required to evaluate the influence of regular judo training in the heat on the risk of exertional heat-related illness.

In the present study, participants commenced exercise in a judo facility at high WBGT of 28.8˚C and 31.6˚C at 09:00 h (AM) and 16:00 h (PM), respectively (Table 1), which

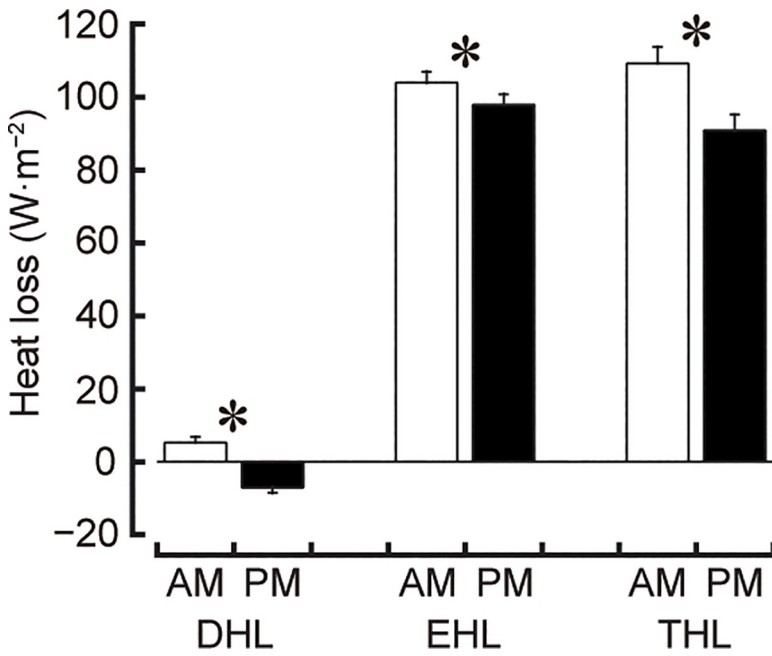

**Fig 3. Responses of dry (DHL), evaporative (EHL) and total (THL) heat losses at the skin during exercise.**
*P<0.001 denotes a difference between AM and PM trials.

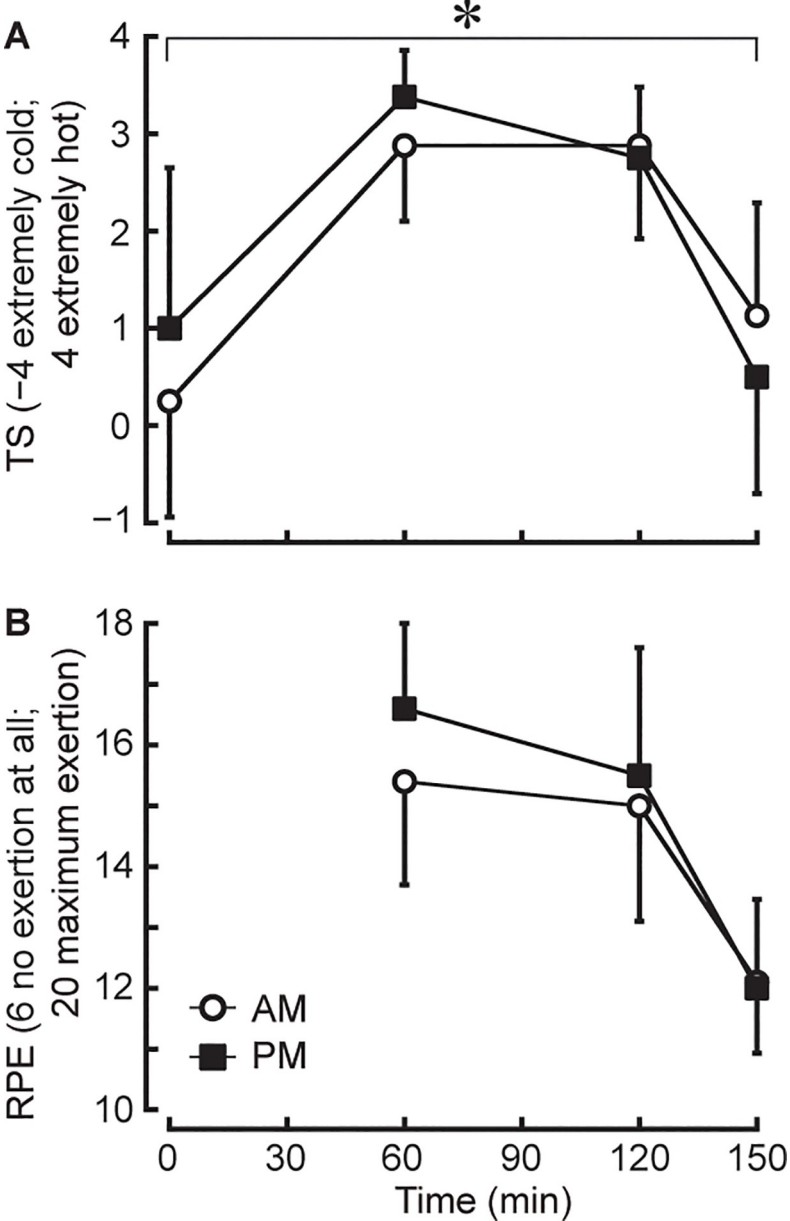

**Fig 4.** Changes in thermal sensation (TS; A) and rating of perceived exertion (RPE; B) during exercise. *$P<0.05$ denotes an interaction between AM and PM trials.

corresponds to an extreme risk category ($\geq 28$°C) for exertional heat-related illness [21]. WBGT kept increasing in AM trial and continued decreasing in PM trial, and WBGT was 29.4°C and 28.8°C at the end of exercise in AM (11:30 h) and PM (18:30 h) trials, respectively (Table 1). These indicate that participants were continuously exposed to heat strain that is regarded as extreme risk for exertional heat-related illness in both trials throughout the training. Indoor heat stress of $T_a$, $T_g$ and $T_r$ also continued to increase during AM trial but they decreased continuously during PM trial that indoor heat stress at the early stages of training was greater in PM than AM trial but the stress at the later stages of training was greater in AM than PM trial (Table 1). Consequently, there was a smaller indoor heat stress with significantly

**Table 2. Spearman's rank correlation coefficianet ($r_s$) between the changes in mean skin temperature (ΔTsk) and heart rate (ΔHR) in each participant and the changes in indoor and outdoor environmental conditions at 60, 120 and 150 min.**

| | Indoor | | | | Outdoor | | | | |
| --- | --- | --- | --- | --- | --- | --- | --- | --- | --- |
| | ΔTa | ΔWBGT | ΔTg | ΔTr | ΔTa | ΔWBGT | ΔTg | ΔTr | ΔSR |
| ΔTsk | | | | | | | | | |
| AM | | | | | | | | | |
| $r_s$ | 0.28 | 0.28 | 0.28 | 0.28 | 0.28 | 0.24 | 0.28 | 0.28 | 0.28 |
| p value | n.s. | n.s. | n.s. | n.s. | n.s. | n.s. | n.s. | n.s. | n.s. |
| PM | | | | | | | | | |
| $r_s$ | 0.74 | 0.74 | 0.74 | 0.74 | 0.08 | 0.74 | 0.74 | 0.74 | 0.74 |
| p value | <0.001 | <0.001 | <0.001 | <0.001 | n.s. | <0.001 | <0.001 | <0.001 | <0.001 |
| ΔHR | | | | | | | | | |
| AM | | | | | | | | | |
| $r_s$ | 0.33 | 0.33 | 0.33 | 0.33 | 0.33 | 0.28 | 0.33 | 0.33 | 0.33 |
| p value | n.s. | n.s. | n.s. | n.s. | n.s. | n.s. | n.s. | n.s. | n.s. |
| PM | | | | | | | | | |
| $r_s$ | 0.77 | 0.77 | 0.77 | 0.77 | 0.52 | 0.77 | 0.77 | 0.77 | 0.77 |
| p value | <0.001 | <0.001 | <0.001 | <0.001 | <0.001 | <0.001 | <0.001 | <0.001 | <0.001 |

Ta, ambient temperature. WBGT, wet-bulb globe temperature. Tg, black globe temperature.

Tr, mean radiant temperature. SR, solar radiation.

lower $T_a$ and $T_g$ during exercise in AM than PM trial (Table 1). These changes were well linked to the changes in outdoor heat stress because $T_a$, WBGT, $T_g$ and $T_r$ in outdoors also kept increasing during AM trial and decreasing during PM trial. This study therefore clearly indicates that indoor heat stress increases with increasing outdoor heat stress in the morning or decreases with decreasing outdoor heat stress in the afternoon in a judo training facility without air conditioning on a clear day in the heat of summer. These changes led to greater outdoor than indoor heat stress during the morning in AM trial and greater indoor than outdoor heat stress during the late afternoon in PM trial. This observation is consistent with the common findings of the architectural studies about the diurnal relationships between indoor and outdoor heat stress in a building during the summer [22].

In agreement with the previous studies during outdoor exercise [1, 2], the current study detected a greater thermoregulatory strain during indoor exercise in AM trial than in PM trial. Average HRmax during exercise was 63±4% in AM trial and 64±5% in PM trial which are corresponding to moderate-intensity exercise [23] and similar to the study that was conducted in 3-h moderate-intensity baseball training [1] but lower than the study that was conducted in 2-h high-intensity football training [2]. These studies reported higher $T_{ty}$ and HR [1, 2] and a higher $T_{sk}$ [1] in the morning than in the afternoon trial. In this study, $T_{sk}$ was higher but heat-loss responses of both DHL and EHL were greater in AM than PM trial (Figs 2B and 3). This means a greater heat-gain during exercise in AM than PM trial. $T_{sk}$ at 60 and 120 min of exercise was exceeding 35°C during both trials which is consistent with [2] but higher than [1] the previous studies. It has been known that $T_{sk}$ of greater than 35°C can evoke the early onset of fatigue in a hot environment [24, 25]. Although the present study did not measure exercise performance, the high $T_{sk}$ may have caused an early decline in judo performance in both trials. Given that the changes in $T_{sk}$ in PM trial was related to the changes in $T_a$, $T_g$, WBGT and $T_r$ in the indoor environment (Table 2), a decrease in indoor heat stress in the late afternoon would attenuate a greater increase in $T_{sk}$ in PM trial and which is in line with Otani et al. [1]. Hence, although indoor heat stress was less in AM than PM trial, the progressive increase in indoor

and outdoor heat stresses in AM trial may have led to a higher $T_{sk}$ in AM trial compared with PM trial.

The studies of Otani et al. [1, 2] observed an interaction with higher $T_{ty}$ and HR at the later stages of training in the morning than the afternoon trial in the heat outdoors under a clear sky, whereas the current study showed only an interaction with no differences at any time point between trials in $T_{ty}$ and HR (Fig 2A and 2C). Given that the current study detected about 3–8°C lower $T_g$ and 12–37°C lower $T_r$ than the studies of Otani et al. [1, 2], the impact of solar radiation may have caused the differences in the diurnal impact on $T_{ty}$ and HR responses between the previous and current studies. In this study, the changes in indoor heat stress as measured by $T_a$, WBGT, $T_g$ and $T_r$, $T_{ty}$ and HR showed similar responses where the values were higher in PM than AM trial at the early stages of training, whilst higher in AM than PM trial at the later stages of training. Therefore, it is possible that the changes in $T_{ty}$ and HR responses are easily influenced by the changes in indoor heat stress during indoor exercise in no air conditioning facility in the heat. This may be responsible for the disagreements in the time-of-day influence on $T_{ty}$ and HR responses between the past [1, 2] and current studies.

Perceived thermal stress (i.e. TS) was greater in PM than AM trial at the early stages of training and in AM than PM trial at the later stages of training, even though perceived fatigue (i.e. RPE) was not different between trials (Fig 4). Previous studies demonstrated that RPE and TS responses were almost similar between the morning and afternoon trials during outdoor exercise, although $T_{ty}$, $T_{sk}$ and HR were higher in the morning than the afternoon trial [1, 2]. In the present study, RPE response was consistent but TS response was inconsistent with the previous studies [1, 2]. These results indicate that perceived fatigue during exercise in the heat may not be influenced by the time-of-day and location (indoor or outdoor) when the same training is performed. Meanwhile, Schlader & Vargas [26] reported that central thermoreceptor activation (i.e. $\Delta T_{core}$) rather than peripheral thermoreceptor activation (i.e. $\Delta$skin temperature) may play a role in perceived thermal stress to exercise in a moderate environment. Although the current study was conducted in a hot environment, statistical analyses revealed the similar changes during exercise between TS and $T_{ty}$ rather than $T_{sk}$. Moreover, given that the changes in TS were also similar to the changes in HR, a higher TS response in PM than AM trial at the early stages of training and in AM than PM trial at the later stages of training would be in accordance with the changes in indoor heat stress as the same responses were observed from $T_{ty}$ and HR. Based on these observations, the chronobiological effect on perceived thermal stress during indoor exercise in the heat may be associated with a combination of the changes in $T_{core}$ and the time-of-day variations in indoor heat stress. Since no studies have systematically examined this effect in any sports including judo, further investigations are required.

The present study is not without limitations. This study used $T_{ty}$ to evaluate $T_{core}$. Nevertheless, previous studies reported that rectal temperature relates to [12] or does not relate to [27] $T_{ty}$ during exercise in the heat. Future research therefore should employ rectal temperature to engage a greater validity and reliability in study regarding the risk of exertional heat-related illness during exercise in the heat of summer. Meanwhile, the current study estimated $f_{cl}$, $R_{cl}$ and $R_{e,cl}$ of judo uniform as 1.23, 0.119 W·(m²·°C)$^{-1}$ and 0.022 W·(m²·kPa)$^{-1}$, respectively, using that of the similar clothing reported. However, we cannot confirm that whether these estimations are within the acceptable difference for true values. This study observed high $T_{sk}$ of greater than 35°C in both trials that might be accompanied by strong heat which could have accumulated inside the judo uniform. To further elucidate the effects of wearing a judo uniform on heat-gain and -loss responses during judo training in the heat, exact clo values for judo ensemble needs to be established. The present study was conducted in a completely sunny condition. This means that outdoor heat stress continued to increase during the

morning and decrease during the afternoon as solar elevation angle rises and falls [28]. That would result in a stable increase of indoor heat stress during AM trial and a stable decrease of indoor heat stress during PM trial in a judo training facility without air conditioning. If the present study was conducted under cloudy conditions, thermoregulatory responses could have been unstable because outdoor and indoor heat stress may not have uniformly increased or decreased. Given this assumption, future study needs to perform the same experiments as the present study under thin or thick cloud conditions.

## Conclusions

We conclude that thermoregulatory strain is greater in the morning from 09:00 h than in the late afternoon from 16:00 h in Japanese high school judokas during 2.5-h regular judo training in a judo training facility without air conditioning on a clear day in the heat of summer. This is attributed to a higher $T_{sk}$ relative to greater DHL and EHL in the morning compared with the late afternoon during exercise, although $T_{ty}$, HR and TS at the early stages of training were higher in the late afternoon than the morning but these at the later stages of training were higher in the morning than in the late afternoon. These findings would be owing to a progressive increase in indoor heat stress with increasing outdoor heat stress in the morning compared with a decrease in indoor heat stress with decreasing outdoor heat stress in the late afternoon when it is a clear day. These observations suggest that judo training in a judo training facility without air conditioning on a clear day in the heat of summer may be at a relatively higher risk for developing exertional heat-related illness in the morning from 09:00 h when the starting WBGT is about 29˚C compared with the late afternoon from 16:00 h. Therefore, we suggest conducting judo training in the afternoon from 16:00 h to minimise the risk of developing an exertional heat-related illness in the heat on a clear day, even if the starting WBGT is about 31˚C. While an exertional heat-related illness is commonly considered as a risk of outdoor activity, athletes and coaches of indoor sports should recognize that activities in indoor facility do not exempt them from heat stress. Proactive heat mitigation strategies applied in other outdoor sports (e.g. adjustment of work to rest ratio, individualized hydration plan) should also be implemented in Judo and other high intensity indoor sports. As WBGT was exceeding 29˚C from the end of AM trial (11:30 h) to the start of PM trial (16:00 h), judo training in such environmental conditions should be avoided to eliminate the mortality and morbidity of heat-related illnesses.

## Supporting information

**S1 Data.**
(DOCX)

## Acknowledgments

The authors thank the participants who donated their time and effort to participate in the present study. The authors also thank Dr. Jos Feys, University of Leuven, for statistical analysis assistance, and Eri Arimoto, Jiei Kusunoki and Ryutaro Tanaka for assistance.

## Author Contributions

**Conceptualization:** Hidenori Otani, Takayuki Goto, Yuki Kobayashi, Minayuki Shirato, Heita Goto, Yuri Hosokawa, Ken Tokizawa, Mitsuharu Kaya.

**Data curation:** Hidenori Otani, Takayuki Goto, Yuki Kobayashi, Mitsuharu Kaya.

**Formal analysis:** Hidenori Otani, Heita Goto, Mitsuharu Kaya.

**Funding acquisition:** Hidenori Otani.

**Investigation:** Hidenori Otani, Takayuki Goto, Yuki Kobayashi, Minayuki Shirato, Heita Goto, Mitsuharu Kaya.

**Methodology:** Hidenori Otani, Takayuki Goto, Yuki Kobayashi, Minayuki Shirato, Heita Goto, Yuri Hosokawa, Ken Tokizawa, Mitsuharu Kaya.

**Project administration:** Hidenori Otani, Takayuki Goto, Yuki Kobayashi, Mitsuharu Kaya.

**Resources:** Hidenori Otani, Takayuki Goto, Yuki Kobayashi, Minayuki Shirato, Heita Goto, Yuri Hosokawa, Ken Tokizawa, Mitsuharu Kaya.

**Software:** Hidenori Otani.

**Supervision:** Hidenori Otani, Takayuki Goto, Yuki Kobayashi, Mitsuharu Kaya.

**Validation:** Hidenori Otani, Takayuki Goto, Yuki Kobayashi, Minayuki Shirato, Heita Goto, Yuri Hosokawa, Ken Tokizawa, Mitsuharu Kaya.

**Visualization:** Hidenori Otani, Takayuki Goto, Yuki Kobayashi, Minayuki Shirato, Heita Goto, Yuri Hosokawa, Ken Tokizawa, Mitsuharu Kaya.

**Writing – original draft:** Hidenori Otani.

**Writing – review & editing:** Hidenori Otani, Takayuki Goto, Yuki Kobayashi, Minayuki Shirato, Heita Goto, Yuri Hosokawa, Ken Tokizawa, Mitsuharu Kaya.

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
