## [Decision Letter · Decision Letter 0]

12 Aug 2020

PONE-D-20-11199

Greater thermoregulatory strain in the morning than late afternoon during judo training in the heat of summer

PLOS ONE

Dear Dr. Otani,

Thank you for submitting your manuscript to PLOS ONE. After careful consideration, we feel that it has merit but does not fully meet PLOS ONE’s publication criteria as it currently stands. Therefore, we invite you to submit a revised version of the manuscript that addresses the points raised during the review process.

In particular, both reviewers commented on the numerous grammatical issues throughout the manuscript. The entire manuscript needs to be reviewed and revised by a native English speaking reviewer or service. Also, please address and answer the questions from both reviewers about the research design and methods and be sure that the conclusion is fully supported by the data presented. Finally, the discussion should be rewritten after the methods and the interpretation of the results are revised as indicated. 

We look forward to receiving your revised manuscript.

Kind regards,

Randy Wayne Bryner, Ed.D.

Academic Editor

PLOS ONE

Journal Requirements:

Reviewers' comments:

Reviewer's Responses to Questions

**Comments to the Author**

1. Is the manuscript technically sound, and do the data support the conclusions?

Reviewer #1: Partly

Reviewer #2: Yes

2. Has the statistical analysis been performed appropriately and rigorously? 

Reviewer #1: No

Reviewer #2: Yes

3. Have the authors made all data underlying the findings in their manuscript fully available?

Reviewer #1: Yes

Reviewer #2: Yes

4. Is the manuscript presented in an intelligible fashion and written in standard English?

Reviewer #1: No

Reviewer #2: No

5. Review Comments to the Author

Reviewer #1: The present paper "Greater thermoregulatory strain in the morning than late afternoon during judo training in the heat of summer" has important methodological problems that lead me to suggest an important review by the authors.

General comments

Reading the paper is difficult, so authors are recommended to use a language review service. I have some doubts about the adequacy of the statistical analyzes used to address the question that arises. In addition, there are certain inaccuracies in the method that must be addressed in order to understand what was done and how it was done. The discussion should be re-writed according to the results obtained after the employment of correct analysis and more about physiological implications for judo training has to be addressed.

Specific comments

Page 9, lines 80-81. This is possible because most judo halls have 81 no air-conditioner owing to its high running costs

Reviewer: please correct English.

Page 9, lines 86. endurance performance

Reviewer: I suggest to add that "judo has high-density efforts" what increases internal temperature (cites).

Page 9, lines 88 and 90. "high body mass index"

Reviewer: the reference is a position statement that cites another position statement that cites references of miners workers. I suggest to seek for a reference on athletes exercising in the heat, and authors should ensure specify if it is BMI or fat mass the cause of worse thermoregulatory capacity.

Page 10, line 121. "The hall is one story,..."

Reviewer: I do not understand what the authors mean.

Page 11, lines 125-126. "...PM trial was conducted first and AM trial was carried out 2 days later..."

Reviewer: why not were the sessions balanced?

Page 11, line 129. "This judo ensemble was 2.5 kg of total weight."

Reviewer: please if the authors consider necessary to report that information, the mean weight and SD should be provided, or minimum and maximum, since judokas had different weight categories are expected to have different sizes of judo is with different extra-weights.

Page 11, lines 132-137. "...including 0.18 clo of short sleeve..."

Reviewer: please define "clo" and it would help to introduce the units expressed in the paragraph to ease the unfamiliar words to be understood by the reader.

Page 11, lines 138-140. "Participants entered a laboratory which is close to the judo hall after a 2 h fast in each trial, other than the ingestion of plain water 30 min before the start of the trial. Upon arrival participants first emptied their bladder and nude body mass was measured to the nearest 10 g"

Reviewer: please re-phrase this sentence.

Page 11, lines 143-155.

Reviewer: the authors must correct the writing style since it is difficult to understand. Furthermore, the authors have to express constantly abbreviations.

Page 12, lines 165. perceived exertion (RPE)

Reviewer: the authors must specify what scale and question were employed to measure perceived exertion/effort, I encourage the review of Halperin & Emmanuel. Sports Med. 2020.

Page 12, lines 166-167. "Skin temperatures and HR were also recorded every 60 min and the end of the training using a thermometer..." - Please re-phrase, it is not clear when the measurement was done. A figure representing the schedule of measurements is encouraged.

Page 12-13. Calculations

Reviewer: I suggest to include this material as supplementary material.

Page 14, line 229-230. "...using Friedman’s two-way ANOVA."

Reviewer: I recommend the employment of the package nparLD (Noguchi et al. Journal of statistical software. 2012) since it allows the study of the interaction (what seems to be the aim of the study) unlike the Friedman's test.

Page 14, lines 231-232."...a two-way (2 [time-of-day] × 3 to 4 [time]) repeated measures ANOVA."

Reviewer: please express the ANOVA factors in a correct way, moreover, the number of measurement does not coincide with the represented in table 1.

Page 14, line 234. "...dependent (indoor vs. outdoor) samples t-test"

Reviewer: please define correctly the dependent variable and if the objective is to compare the time of the day and place of measurement (indoor vs outdoor), as it seems, an ANOVA test is more suitable.

Page 15, lines 247-257.

Reviewer: I really do not understand how the authors made the statistical analysis and I have my concerns about the interpretation of the results according to the statistical analysis described in the methods section.

Page 16, lines 273-274. Average HR during exercise was not different between trials (AM 63±4% HRmax, PM 64±5% 274 HRmax; P=0.680).

Reviewer: it seems that HR is from the entire session, an analysis of the more demanding task would clarify better the internal load imposed on the judokas and it could facilitate the interpretation of the possible effects of time of the day and indoor temperature on the cardiovascular effort of the athletes. An important point to consider is that likely the more physically demanding part of the sessions was performed in the part of the session with higher indoor temperature in AM and in the part of the sessions with lower indoor temperature in PM. Thus, an analysis of HR measured in relation to the task performed is required to interpret the data.

Page 17, lines 311-314. "A novel finding in this study is that there is a greater thermoregulatory strain in the morning from 09:00 h than in the late afternoon from 16:00 h during 2.5-h regular judo training in no air conditioning judo hall in the heat of summer irrespective of environmental heat stress."

Reviewer: the conclusion of the study has bias because the evolution of temperature has to be analysed regarding with the demands of the judo task, since the cumulative effect of the increase of temperature and higher efforts can influence the physiological effects on judocas.

Page 18, lines 364-365. "Average HRmax during exercise was 63±4% in AM trial and 64±5% in PM trial which are corresponding to moderate-intensity exercise [27] and similar to [1] but lower than [2] the previous..."

Reviewer: to compare data between studies it would be necessary have into account the task performed by the athletes.

Reviewer #2: The manuscript presents interesting data from apparently well-conducted investigation examining the influence of AM versus PM training sessions; however, the document needs substantial language editing prior to further consideration for publication. Some additional comments and suggestions are provided below.

Consider an alternative to “no air conditioning judo hall” throughout the paper; perhaps “judo training facility without air conditioning.”

The following sentence used in the abstract and elsewhere is confusing: “showing relatively higher responses in these variables in PM than AM and in AM than PM at the early and the later stages of training, respectively.” Perhaps the following would be more clear: “showing relatively higher responses in these variables in the PM compared to the AM during the early stages of training and in the AM compared to the PM during the later stages of training.”

Some of the following additional relevant details for the participants and environment may be useful:

What is the training age or the number of years of training for the partcipants?

Did they typically engage in the type of judo training examined during the study in both the AM and PM or was one time reserved for other types of training (running, lifting weights, newaza, etc.)?

How much time had elapsed since the previous training session?

Did the athletes start the training in a dry judogi?

Regarding the different finding for TS and RPE, is it possible that the judo athletes self-regulate to adjust to an expected or typical RPE while still acknowledging differences in TS?

6. PLOS authors have the option to publish the peer review history of their article (what does this mean?). If published, this will include your full peer review and any attached files.

Reviewer #1: No

Reviewer #2: No

---

## [Author Response · Author response to Decision Letter 0]

1 Sep 2020

Response to Reviewer 1

We are grateful to reviewer for the critical comments and useful suggestions that have helped us to improve our manuscript. As indicated in the responses that follow, we have taken all these comments and suggestions into account in the revised version of our manuscript.

Comment#1:

General comments

Reading the paper is difficult, so authors are recommended to use a language review service. I have some doubts about the adequacy of the statistical analyzes used to address the question that arises. In addition, there are certain inaccuracies in the method that must be addressed in order to understand what was done and how it was done. The discussion should be re-writed according to the results obtained after the employment of correct analysis and more about physiological implications for judo training has to be addressed.

Response to Comment#1:

Thank you for these comments. This paper has been proofread by a native English speaker again. The changes are shown in red in the revised version of text (clean copy). We hope that our comments below are relevant to the response to the reviewer’s comments.

Comment#2:

Specific comments

Page 9, lines 80-81. This is possible because most judo halls have 81 no air-conditioner owing to its high running costs

Reviewer: please correct English.

Response to Comment#2:

We agree with this comment. In relation to another reviewer’s comment, we would like to replace this sentence “This is possibly because most judo halls have no air-conditioner owing to its high running costs.” with “This is possibly due to luck of air conditioning in most judo facilities owing to its high running costs.”. 

Comment#3:

Page 9, lines 86. endurance performance

Reviewer: I suggest to add that "judo has high-density efforts" what increases internal temperature (cites).

Response to Comment#3:

Thank you for this suggestion. We have added “has high-density efforts and” to this sentence in line 85 in the revised version of text (clean copy).

Comment#4:

Page 9, lines 88 and 90. "high body mass index"

Reviewer: the reference is a position statement that cites another position statement that cites references of miners workers. I suggest to seek for a reference on athletes exercising in the heat, and authors should ensure specify if it is BMI or fat mass the cause of worse thermoregulatory capacity.

Response to Comment#4:

This reference (Casa et al. J Athl Train. 2015) cited two references about the effects of the risk in high body mass index on heat-related illnesses. One is the study of Cleary (J Sport Rehabil. 2007;16(3):204–214), the other is the study of Chung and Pin (Mil Med. 1996;161(12):739–742). We are afraid that we confirmed both studies analysing the data during exercise. Therefore, we would like to use the reference of Casa et al. as it is.

Comment#5:

Page 10, line 121. "The hall is one story,..."

Reviewer: I do not understand what the authors mean.

Response to Comment#5:

In relation to another reviewer’s comment, we would like to replace this sentence “The hall is one story,…” with “The judo facility is a one-story building with a floor space of 225 m2 (15 m × 15m).”.　

Comment#6:

Page 11, lines 125-126. "...PM trial was conducted first and AM trial was carried out 2 days later..."

Reviewer: why not were the sessions balanced?

Response to Comment#6:

We understand the reviewer’s comment. However, to keep constant environmental conditions during the trial, we had to carry out each trial in all participants together. Therefore, we had no choice but to carry out first either AM trial or PM trial.

Comment#7:

Page 11, line 129. "This judo ensemble was 2.5 kg of total weight."

Reviewer: please if the authors consider necessary to report that information, the mean weight and SD should be provided, or minimum and maximum, since judokas had different weight categories are expected to have different sizes of judo is with different extra-weights.

Response to Comment#7:

Thank you for this suggestion. We have replaced this sentence “This judo ensemble was 2.5 kg of total weight.” with “The judo ensemble was 2.5±0.1kg of total weight.”.

Comment#8:

Page 11, lines 132-137. "...including 0.18 clo of short sleeve..."

Reviewer: please define "clo" and it would help to introduce the units expressed in the paragraph to ease the unfamiliar words to be understood by the reader.

Response to Comment#8:

We agree with this comment. We have added the following sentence “A clo is a unit of thermal insulation for clothing: one clo can be defined as the amount of insulation that allows the transfer of 1 W·m−2 with a temperature gradient of 0.155°C between two surfaces (0.18°C·m2·h·kcal−1).” to line 136-138 in the revised version of text (clean copy).

Comment#9:

Page 11, lines 138-140. "Participants entered a laboratory which is close to the judo hall after a 2 h fast in each trial, other than the ingestion of plain water 30 min before the start of the trial. Upon arrival participants first emptied their bladder and nude body mass was measured to the nearest 10 g"

Reviewer: please re-phrase this sentence.

Response to Comment#9:

We agree with this comment. We have changed these sentences to “Participants entered a laboratory which was close to the judo facility after a 2 h fast in each trial with the exception of plain water, which was allowed until 30 min before the start of the trial. Upon arrival, participants first emptied their bladder and thereafter nude body mass was measured to the nearest 10 g”.

Comment#10:

Page 11, lines 143-155.

Reviewer: the authors must correct the writing style since it is difficult to understand. Furthermore, the authors have to express constantly abbreviations.

Response to Comment#10:

We agree with this comment. We have changed these sentences to

“Surface skin temperature thermistor probes (ITP082-25, Nikkiso-Therm Co., Ltd., Musashino, Tokyo, Japan) were attached to four sites (chest, upper arm, thigh and calf) under the clothing without preventing range of motion. A weighted average of chest (0.3), upper arm (0.3), thigh (0.2) and calf (0.2) skin temperatures was used to calculate mean skin temperature (Tsk) [10]. Gastrointestinal thermometry has been shown to be a valuable device for core temperature (Tcore) assessment in the field and athletics settings [11]. However, the current study measured an infrared tympanic temperature (Tty) to estimate Tcore due to the restriction from pharmaceutical affairs law in Japan using gastrointestinal thermometry. Tty was measured using an infrared tympanic thermometer (GeniusTM 2, Covidien, Mansfield, MA, USA). In each measurement, two consecutive readings were obtained. All Tty measurements were taken by a single operator, using the recommended technique [12]. To avoid the increased effects of increasing Ta on Tty in the heat [13], the thermometer was stored inside a cooling box during the trial. The temperature inside this box was maintained by ice packs at about 25°C. Thermal sensation (TS) was measured using a 9-point scale [14]. All pre-exercise measurements were carried out in the laboratory in a temperate environment (25-27°C Ta) because prior heat stress exposure may increase thermoregulatory strain during subsequent exercise-heat stress in the morning than in the afternoon [15].”

 In accordance with this change, we would like to add the following sentences to line 3-4 in supporting information (S1 Methods-Calculations) in relation to Comment#13, “Tsk was calculated using the following equation [10]: Tsk = 0.3×chest + 0.3×upper arm + 0.2×thigh + 0.2×calf [ºC].”.

Comment#11:

Page 12, lines 165. perceived exertion (RPE)

Reviewer: the authors must specify what scale and question were employed to measure perceived exertion/effort, I encourage the review of Halperin & Emmanuel. Sports Med. 2020.

Response to Comment#11:

Thank you for this comment and introducing a good manuscript. In accordance with this, we have replaced this sentence “During the sessions, Tty, TS and rating of perceived exertion (RPE) [17] were assessed every 60 min and the end of the training.” with “During the sessions, Tty and TS were assessed every 60 min and the end of the training. To determine whole-body perception of effort, rating of perceived exertion (RPE) was assessed every 60 min and the end of the training using the 6-20 RPE scale [17].”. 

Comment#12:

Page 12, lines 166-167. "Skin temperatures and HR were also recorded every 60 min and the end of the training using a thermometer..." - Please re-phrase, it is not clear when the measurement was done. A figure representing the schedule of measurements is encouraged.

Response to Comment#12:

We agree with the reviewer’s comment. we have replaced the following sentences “Skin temperatures and HR were also recorded every 60 min and the end of the training using a thermometer (N543R, Nikkiso-Therm Co., Ltd., Musashino, Tokyo, Japan) and HR monitor (A370, Polar Electro, Kempele, Finland), respectively.” with “Skin temperatures (thermometer N543R, Nikkiso-Therm Co., Ltd., Musashino, Tokyo, Japan) and HR (HR monitor A370, Polar Electro, Kempele, Finland) were also recorded every 60 min and the end of the training (Fig 1)”.

Moreover, in relation to this comment and another reviewer’s comment#4, we would like to add the experimental protocol as Figure 1. Therefore, we would like to replace “Fig 1” with “Fig 2”, “Fig 2” with “Fig 3”, and “Fig 3” with “Fig 4” throughout the text. Moreover, we would like to add the following figure legends, “Fig 1. Schematic representation of the experimental protocol. Tty, infrared tympanic temperature; HR, heart rate; TS, thermal sensation; RPE, rating of perceived exertion.”.

Comment#13:

Page 12-13. Calculations

Reviewer: I suggest to include this material as supplementary material.

Response to Comment#13:

We agree with the reviewer’s suggestion. We have transferred the sentences from line 185 to line 223 to the supplementary material. In accordance with this, we have added the following sentence to line 197-199 in the revised version of text (clean copy), “The equations of Tsk, mean radiant temperature (Tr), dry or sensible heat loss (DHL), evaporative heat loss (EHL), total heat loss (THL), absolute humidity and total sweat loss are included in supporting information (S1 Methods-Calculations).”.

Comment#14:

Page 14, line 229-230. "...using Friedman’s two-way ANOVA."

Reviewer: I recommend the employment of the package nparLD (Noguchi et al. Journal of statistical software. 2012) since it allows the study of the interaction (what seems to be the aim of the study) unlike the Friedman's test.

Response to Comment#14:

We appreciate this suggestion. In accordance with this, we have re-calculated thermal sensation (TS) using the R package nparLD by helping Dr. Jos Feys, University of Leuven. However, the results of TS were not substantially different that in the original version of the text. In relation to this, we have changed the following points:

1. Removed the following sentence from line 226-227 in the original version of text, “The IBM SPSS (version 21; IBM Corp., Armonk, N.Y., USA) was used for all statistical analyses.”

2. Replaced the following sentence in line 229-230 in the original version of text, “Non-parametric data (TS) were analysed using Friedman’s two-way ANOVA.” with “Non-parametric data (TS) were analysed using R. TS was analysed using a two-way (time-of-day [two levels, i.e., AM and PM] × time [four levels, i.e., 0, 60, 120 and 150 min]) repeated measures ANOVA with the R package nparLD. Pair-wise differences between trials were evaluated using the Kruskal-Wallis test with the Wilcoxon rank sum test.”

3. Added the following sentence to line 205-206 in the revised version of text (clean copy), “statistical analyses of data were done in the IBM SPSS (version 21; IBM Corp., Armonk, N.Y., USA).”

4. Replaced the following sentence, line 230-231 in the original version of text, “, data collected once per trial were analysed using a one-way repeated measures ANOVA,” with “Data collected once per trial were analysed using a one-way repeated measures ANOVA,”

5. Replaced the following sentences, line 290-292 in the original version of text, “A trial by time interaction effect was shown for TS (P<0.05; 1−β=0.82), but post hoc analysis revealed no difference at any time point between trials (all P>0.05; Fig 3A). Also, there was no main effect of trial in TS (P=0.381).” with “A trial by time interaction effect was shown for TS (P<0.05), but post hoc analysis revealed no difference at any time point between trials (all P>0.05; Fig 4A). Also, there was no main effect of trial in TS (P=0.137).”.

6. Replaced the following sentence to Acknowledgements, line 459 in the original version of text, “The authors also thank, Eri Arimoto, Jiei Kusunoki and Ryutaro Tanaka for assistance.” with “The authors also thank Dr. Jos Feys, University of Leuven, for statistical analysis assistance, and Eri Arimoto, Jiei Kusunoki and Ryutaro Tanaka for assistance.”.

Comment#15:

Page 14, lines 231-232."...a two-way (2 [time-of-day] × 3 to 4 [time]) repeated measures ANOVA."

Reviewer: please express the ANOVA factors in a correct way, moreover, the number of measurement does not coincide with the represented in table 1.

Response to Comment#15:

We agree with this comment. We have replaced this sentence “using a two-way (2 [time-of-day] × 3 to 4 [time]) repeated measures ANOVA.” with “using a two-way (time-of-day [two levels, i.e., AM and PM] × time [three or four levels, i.e., 0, 60, 120 and 150 min]) repeated measures ANOVA.”.

 The sampling interval of each environmental conditions data were expressed in line 176-182 in the original version of text. We would like to add the sampling interval of environmental conditions in Figure 1.

Comment#16:

Page 14, line 234. "...dependent (indoor vs. outdoor) samples t-test"

Reviewer: please define correctly the dependent variable and if the objective is to compare the time of the day and place of measurement (indoor vs outdoor), as it seems, an ANOVA test is more suitable.

Response to Comment#16:

We agree with the reviewer’s comment. We have replaced the following phrase, line 234-235, “...dependent (indoor vs. outdoor) samples t-test” with “…dependent (indoor AM vs. outdoor AM; indoor PM vs. outdoor PM) samples t-test”.

Meanwhile, given that indoor and outdoor environmental variables were measured at the same time and collected once per each sampling point in this study, these variables are need to analyse the dependent samples t-test between indoor and outdoor environments. An ANOVA test is suitable when there are more than three groups but is inappropriate between the two groups. By the way, when two groups are compared, the results of statistical analysis (the P value) are completely the same between the dependent sample t-test and the repeated measures ANOVA.

Comment#17:

Page 15, lines 247-257.

Reviewer: I really do not understand how the authors made the statistical analysis and I have my concerns about the interpretation of the results according to the statistical analysis described in the methods section.

Response to Comment#17:

We hope that our comments in Response to Comment#16 are relevant to the response to this comment.

Comment#18:

Page 16, lines 273-274. Average HR during exercise was not different between trials (AM 63±4% HRmax, PM 64±5% 274 HRmax; P=0.680).

Reviewer: it seems that HR is from the entire session, an analysis of the more demanding task would clarify better the internal load imposed on the judokas and it could facilitate the interpretation of the possible effects of time of the day and indoor temperature on the cardiovascular effort of the athletes. An important point to consider is that likely the more physically demanding part of the sessions was performed in the part of the session with higher indoor temperature in AM and in the part of the sessions with lower indoor temperature in PM. Thus, an analysis of HR measured in relation to the task performed is required to interpret the data.

Response to Comment#18:

We agree with this comment. We would like to add the following sentences to line 250-254 in the revised version of text (clean copy), “The percentage of HRmax (% HRmax) at 60, 120 and 150 min was 67±6%, 67±6% and 55±3% in AM trial and 72±7%, 63±4% and 57±7% in PM trial. There was a trial by time interaction effect for % HRmax (P<0.05; 1−β=0.87), but post hoc analysis revealed no difference at any time point between trials (all P>0.05).”.

Comment#19:

Page 17, lines 311-314. "A novel finding in this study is that there is a greater thermoregulatory strain in the morning from 09:00 h than in the late afternoon from 16:00 h during 2.5-h regular judo training in no air conditioning judo hall in the heat of summer irrespective of environmental heat stress."

Reviewer: the conclusion of the study has bias because the evolution of temperature has to be analysed regarding with the demands of the judo task, since the cumulative effect of the increase of temperature and higher efforts can influence the physiological effects on judocas.

Response to Comment#19:

We agree with this comment. We have changed this sentence to “A novel finding in this study is that there is a greater thermoregulatory strain in the morning from 09:00 h than in the late afternoon from 16:00 h during 2.5-h regular judo training in judo training facility without air conditioning in the heat of summer. This finding is associated with the progressive increase in indoor and outdoor heat stresses in the morning compared with the progressive decrease in indoor and outdoor heat stresses in the late afternoon.”

Comment#20:

Page 18, lines 364-365. "Average HRmax during exercise was 63±4% in AM trial and 64±5% in PM trial which are corresponding to moderate-intensity exercise [27] and similar to [1] but lower than [2] the previous..."

Reviewer: to compare data between studies it would be necessary have into account the task performed by the athletes.

Response to Comment#20:

We appreciate this suggestion. In accordance with this, we have changed this sentence to “Average HRmax during exercise was 63±4% in AM trial and 64±5% in PM trial which are corresponding to moderate-intensity exercise [27] and similar to the study that was conducted in 3-h moderate-intensity baseball training [1] but lower than the study that was conducted in 2-h high-intensity football training [2]”.

We are most grateful for your comments on our manuscript. We trust that the revised version of our manuscript is suitable for publication.

Response to Reviewer 2

We are grateful to reviewer for the critical comments and useful suggestions that have helped us to improve our manuscript. As indicated in the responses that follow, we have taken all these comments and suggestions into account in the revised version of our manuscript.

Comment#1:

Consider an alternative to “no air conditioning judo hall” throughout the paper; perhaps “judo training facility without air conditioning.”

Response to Comment#1:

We appreciate this suggestion. In accordance with this, we have replaced “no air conditioning judo hall” with “judo training facility without air conditioning” throughout the paper.

In relation to this, we have replaced “hall/halls” with “facility/facilities” throughout the paper.

Comment#2:

The following sentence used in the abstract and elsewhere is confusing: “showing relatively higher responses in these variables in PM than AM and in AM than PM at the early and the later stages of training, respectively.” Perhaps the following would be more clear: “showing relatively higher responses in these variables in the PM compared to the AM during the early stages of training and in the AM compared to the PM during the later stages of training.”

Response to Comment#2:

Thank you for this comment. We agree with this comment. We have replaced “showing relatively higher responses in these variables in PM than AM and in AM than PM at the early and the later stages of training, respectively” with “showing relatively higher responses in these variables in PM compared to AM during the early stages of training and in AM compared to PM during the later stages of training” in the Abstract.

Comment#3:

Some of the following additional relevant details for the participants and environment may be useful:

What is the training age or the number of years of training for the partcipants?

Response to Comment#3:

We agree with this comment. In accordance with this, we have added the following phrase “, years of training 6±2 y” to line 109 in the revised version of text (clean copy).

Comment#4:

Did they typically engage in the type of judo training examined during the study in both the AM and PM or was one time reserved for other types of training (running, lifting weights, newaza, etc.)?

Response to Comment#4:

We are sorry that our explanation about the type of judo training (Line 161-165 in the original version of text) was a little confusing. To clarify this and in relation to another reviewer’s comment#12, we would like to add the experimental protocol as Figure 1. In relation to this change, we would like to replace “Fig 1” with “Fig 2”, “Fig 2” with “Fig 3”, and “Fig 3” with “Fig 4” throughout the text. Moreover, we would like to add the following figure legends, “Fig 1. Schematic representation of the experimental protocol. Tty, infrared tympanic temperature; HR, heart rate; TS, thermal sensation; RPE, rating of perceived exertion.”.

Comment#5:

How much time had elapsed since the previous training session?

Response to Comment#5:

We agree with this comment. In accordance with this, we have added the following sentence to line 127-129 in the revised version of text (clean copy), “A normal training session took place two days before the first trial (PM trial) but no exercise was permitted during the 24 h prior to the trials.”.

Comment#6:

Did the athletes start the training in a dry judogi?

Response to Comment#6:

Yes, exactly. We have added the following sentence to line 163-164 in the revised version of text (clean copy), “Participants started the sessions in a dry judo uniform.”.

Comment#7:

Regarding the different finding for TS and RPE, is it possible that the judo athletes self-regulate to adjust to an expected or typical RPE while still acknowledging differences in TS?

Response to Comment#7:

We think that no differences in RPE between AM and PM trials may be accompanied by the same content of training in both trials, regardless of the differences in TS. However, as the reviewer have pointed out, it was possible that participants self-regulated their exercise intensity to prevent excessive fatigue, especially in the early stages of training. 

We are most grateful for your comments on our manuscript. We trust that the revised version of our manuscript is suitable for publication.

---

## [Decision Letter · Decision Letter 1]

13 Oct 2020

PONE-D-20-11199R1

Greater thermoregulatory strain in the morning than late afternoon during judo training in the heat of summer

PLOS ONE

Dear Dr. Otani,

Thank you for submitting your manuscript to PLOS ONE. After careful consideration, we feel that it has merit but does not fully meet PLOS ONE’s publication criteria as it currently stands. Therefore, we invite you to submit a revised version of the manuscript that addresses the points raised during the review process.

Please carefully review the manuscript and correct the few grammatical errors.

Please specify the version of R.

Please define thoroughly HRmax. 

We look forward to receiving your revised manuscript.

Kind regards,

Randy Wayne Bryner, Ed.D.

Academic Editor

PLOS ONE

Reviewers' comments:

Reviewer's Responses to Questions

**Comments to the Author**

1. If the authors have adequately addressed your comments raised in a previous round of review and you feel that this manuscript is now acceptable for publication, you may indicate that here to bypass the “Comments to the Author” section, enter your conflict of interest statement in the “Confidential to Editor” section, and submit your "Accept" recommendation.

Reviewer #1: All comments have been addressed

Reviewer #2: (No Response)

2. Is the manuscript technically sound, and do the data support the conclusions?

Reviewer #1: Yes

Reviewer #2: Yes

3. Has the statistical analysis been performed appropriately and rigorously? 

Reviewer #1: Yes

Reviewer #2: Yes

4. Have the authors made all data underlying the findings in their manuscript fully available?

Reviewer #1: Yes

Reviewer #2: Yes

5. Is the manuscript presented in an intelligible fashion and written in standard English?

Reviewer #1: Yes

Reviewer #2: No

6. Review Comments to the Author

Reviewer #1: I would like to congratulate the authors since they have made a great work adressing the reviwers comments and rise the paper to the standards to be published in Plosone.

Reviewer #2: Minor grammatical errors still exist throughout the text that should be easy enough to address with a thorough review.

Please specify the version of R.

An operational definition of HRmax is needed to provide context for the current results.

7. PLOS authors have the option to publish the peer review history of their article (what does this mean?). If published, this will include your full peer review and any attached files.

Reviewer #1: **Yes: **Eduardo Carballeira

Reviewer #2: No

---

## [Author Response · Author response to Decision Letter 1]

20 Oct 2020

Response to Reviewer 1

We are grateful to reviewer for the critical comments and useful suggestions that have helped us to improve our manuscript. 

We are afraid that our statistical adviser (Dr. Jos Feys, University of Leuven) has pointed out that the Tukey multiple comparison tests rather than the Kruskal-Wallis test with the Wilcoxon rank sum test are a practical and powerful approach to multiple testing in this TS data. The results of statistical analysis in the TS data were not different the present results when using the Tukey multiple comparison tests. Therefore, we would like to replace the following sentence, line 205-206,

“Pair-wise differences between trials were evaluated using the Kruskal-Wallis test with the Wilcoxon rank sum test.”

with

“Pair-wise differences between trials were evaluated using the Tukey multiple comparison tests.”

We are most grateful for your comments on our manuscript. We trust that the second revised version of our manuscript is suitable for publication.

Response to Reviewer 2

We are grateful to reviewer for the critical comments and useful suggestions that have helped us to improve our manuscript. As indicated in the responses that follow, we have taken all these comments and suggestions into account in the second revised version of our manuscript.

Comment#1:

Minor grammatical errors still exist throughout the text that should be easy enough to address with a

thorough review.

Response to Comment#1:

In accordance with the reviewer’s comment, this paper has been re-proofread by a native English speaker. The changes are shown in red in the revised version of text. 

Comment#2:

Please specify the version of R.

Response to Comment#2:

We agree with this comment. To add the version of R, we have replaced the following sentences, line 203-205,

“Non-parametric data (TS) were analysed using R. TS was analysed using a two-way (time-of-day [two levels, i.e., AM and PM] × time [four levels, i.e., 0, 60, 120 and 150 min]) repeated measures ANOVA with the R package nparLD”

with

“Non-parametric data (TS) were analysed using R (version 4.0.2). TS was analysed using a two-way (time-of-day [two levels, i.e., AM and PM] × time [four levels, i.e., 0, 60, 120 and 150 min]) repeated measures ANOVA with the R package nparLD (version 2.1) for the LD-F2 design.”.

Comment#3:

An operational definition of HRm ax is needed to provide context for the current results.

Response to Comment#3:

We agree with this comment. In relation to this comment, we have changed the following points,

1) line 196-198: Replaced the following sentence

“The equations of Tsk, mean radiant temperature (Tr), dry or sensible heat loss (DHL), evaporative heat loss (EHL), total heat loss (THL), absolute humidity and total sweat loss are included in supporting information (S1 Methods-Calculations).”

with

“The equations of Tsk, mean radiant temperature (Tr), dry or sensible heat loss (DHL), evaporative heat loss (EHL), total heat loss (THL), absolute humidity, total sweat loss and maximal HR (HRmax) are included in supporting information (S1 Methods-Calculations).”.

2) S1 Methods-Calculations, the bottom of the text: Add the following sentence

“HRmax was calculated by subtracting the age from 220.”

We are afraid that our statistical adviser (Dr. Jos Feys, University of Leuven) has pointed out that the Tukey multiple comparison tests rather than the Kruskal-Wallis test with the Wilcoxon rank sum test are a practical and powerful approach to multiple testing in this TS data. The results of statistical analysis in the TS data were not different the present results when using the Tukey multiple comparison tests. Therefore, we would like to replace the following sentence, line 205-206,

“Pair-wise differences between trials were evaluated using the Kruskal-Wallis test with the Wilcoxon rank sum test.”

with

“Pair-wise differences between trials were evaluated using the Tukey multiple comparison tests.”

We are most grateful for your comments on our manuscript. We trust that the second revised version of our manuscript is suitable for publication.

---

## [Decision Letter · Decision Letter 2]

12 Nov 2020

Greater thermoregulatory strain in the morning than late afternoon during judo training in the heat of summer

PONE-D-20-11199R2

Dear Dr. Otani,

We’re pleased to inform you that your manuscript has been judged scientifically suitable for publication and will be formally accepted for publication once it meets all outstanding technical requirements.

Kind regards,

Randy Wayne Bryner, Ed.D.

Academic Editor

PLOS ONE

Additional Editor Comments (optional):

Reviewers' comments:

Reviewer's Responses to Questions

**Comments to the Author**

1. If the authors have adequately addressed your comments raised in a previous round of review and you feel that this manuscript is now acceptable for publication, you may indicate that here to bypass the “Comments to the Author” section, enter your conflict of interest statement in the “Confidential to Editor” section, and submit your "Accept" recommendation.

Reviewer #2: (No Response)

2. Is the manuscript technically sound, and do the data support the conclusions?

Reviewer #2: Yes

3. Has the statistical analysis been performed appropriately and rigorously? 

Reviewer #2: Yes

4. Have the authors made all data underlying the findings in their manuscript fully available?

Reviewer #2: Yes

5. Is the manuscript presented in an intelligible fashion and written in standard English?

Reviewer #2: No

6. Review Comments to the Author

Reviewer #2: Line 199 - consider using "age-predicted maximal HR"

The revision to "a regular judo training" from "regular judo training" is unnecessary.

7. PLOS authors have the option to publish the peer review history of their article (what does this mean?). If published, this will include your full peer review and any attached files.

Reviewer #2: No

---

## [Editor Report · Acceptance letter]

16 Nov 2020

PONE-D-20-11199R2 

Greater thermoregulatory strain in the morning than late afternoon during judo training in the heat of summer  

Dear Dr. Otani:

I'm pleased to inform you that your manuscript has been deemed suitable for publication in PLOS ONE. Congratulations! Your manuscript is now with our production department. 

Kind regards, 

on behalf of

Dr. Randy Wayne Bryner 

Academic Editor

PLOS ONE